# Evaluation of the uptake and delivery of the NHS Health Check programme in England, using primary care data from 9.5 million people: a cross-sectional study

Riyaz Patel ,[1] Sharmani Barnard,[2] Katherine Thompson,[2] Catherine Lagord,[2] Emma Clegg,[2] Robert Worrall,[3] Tim Evans,[2] Slade Carter,[2] Julian Flowers,[2] Dave Roberts,[3] Michaela Nuttall,[2] Nilesh J Samani,[4,5] John Robson,[6] Matt Kearney,[7] John Deanfield,[1] Jamie Waterall[2]

► Prepublication history and supplemental material for this paper is available online. To view these files, please visit the journal online (http://dx.doi.org/10.1136/bmjopen-2020-042963).

RP and SB are joint first authors. JD and JW are joint senior authors.

For numbered affiliations see end of article.

**Correspondence to**
Dr Riyaz Patel;
riyaz.patel@ucl.ac.uk

## ABSTRACT

**Objectives** To describe the uptake and outputs of the National Health Service Health Check (NHSHC) programme in England.

**Design** Observational study.

**Setting** National primary care data extracted directly by NHS Digital from 90% of general practices (GP) in England.

**Participants** Individuals aged 40–74 years, invited to or completing a NHSHC between 2012 and 2017, defined using primary care Read codes.

**Intervention** The NHSHC, a structured assessment of non-communicable disease risk factors and 10-year cardiovascular disease (CVD) risk, with recommendations for behavioural change support and therapeutic interventions.

**Results** During the 5-year cycle, 9 694 979 individuals were offered an NHSHC and 5 102 758 (52.6%) took up the offer. There was geographical variation in uptake between local authorities across England ranging from 25.1% to 84.7%. Invitation methods changed over time to incorporate greater digitalisation, opportunistic delivery and delivery by third-party providers.

The population offered an NHSHC resembled the English population in ethnicity and deprivation characteristics. Attendees were more likely to be older and women, but were similar in terms of ethnicity and deprivation, compared with non-attendees. Among attendees, risk factor prevalence reflected population survey estimates for England. Where a CVD risk score was documented, 25.9% had a 10-year CVD risk ≥10%, of which 20.3% were prescribed a statin. Advice, information and referrals were coded as delivered to over 2.5 million individuals identified to have risk factors.

**Conclusion** This national analysis of the NHSHC programme, using primary care data from over 9.5 million individuals offered a check, reveals an uptake rate of over 50% and no significant evidence of inequity by ethnicity or deprivation. To maximise the anticipated value of the NHSHC, we suggest continued action is needed to invite more eligible people for a check, reduce geographical variation in uptake, prioritise engagement with non-

### Strengths and limitations of this study

► A comprehensive national-level snapshot of National Health Service Health Check (NHSHC) programme, derived from primary care records, and which underpins the recently released NHSHC data dashboard.

► Academic and public health collaboration with full access to half a billion records for over 9.5 million people offered an NHSHC between 2012 and 2017.

► This first data analysis reports on elements relating to uptake, implementation, process and delivery of NHSHCs, the sociodemographic and risk factor profile of both those who did and did not attend a check and rates of advice, referrals and statin prescriptions delivered as part of the check.

► The data were restricted to people with an NHSHC activity code, and thus we were unable to quantify the full eligible population to determine coverage and the gap in programme reach.

► Missing data and varying volume of completeness of risk factor measures limit comparisons between attendees and non-attendees.

attendees and promote greater use of evidence-based interventions especially where risk is identified.

## INTRODUCTION

Cardiovascular disease (CVD) remains a major public health priority in England.[1] To address this, the government introduced an ambitious programme of vascular checks in 2009, for people aged 40–74, delivered by England's National Health Service (NHS).[2] NHS Health Checks (NHSHCs) sought to address the key risk factors driving the health and economic burden from vascular disease,[3] with early

modelling suggesting that each year NHSHCs would prevent 9500 heart attacks and strokes, 4000 new cases of diabetes and identify at least 25 000 people with existing undiagnosed diabetes or kidney disease before they developed complications.[2] [4] Furthermore, with the same vascular risk factors increasingly recognised as contributing to other conditions like dementia, preventable cancers and liver disease,[3] the programme has assumed an even greater importance in the prevention of non-communicable diseases (NCD).[5–7]

Over a decade on, the NHSHC is now an embedded systematic and nationwide detailed risk assessment, awareness and management programme in England. Since 2013, following legislation, local authorities have a statutory obligation to make provision for all eligible people to have an NHSHC every 5 years.[8] However, concerns have been raised that delivery and practical implementation of such a programme presents a paradoxical risk of increasing health inequality if implemented in a way which does not systematically prioritise equity of access, outputs and outcomes. Furthermore, the absence of convincing randomised clinical trial evidence about the effectiveness of such programmes has further prompted ongoing scrutiny and questions around its delivery, uptake, impact and cost-effectiveness.[9]

In response, the number of studies evaluating the delivery and impact of the NHSHC continue to grow but have shown variable results.[10] This may be a result of heterogeneity in programme delivery, small sample sizes, use of national data before NHSHCs were passed into law or variation in local coding practices. In addition, some studies have drawn conclusions from analyses of the Clinical Practice Research Datalink or QResearch databases,[11] which although a representative and important primary care research resource, are limited by being restricted to volunteer practices using specific electronic health record systems with some under-representation in Northern England.[11] [12]

To overcome some of these difficulties and provide a contemporaneous overview of the NHSHC programme in England, we sought to analyse the largest NHSHC national primary care dataset to be extracted to date, drawing on data for almost 10 million individuals and half a billion records, specifically extracted for this purpose and one which underpins the recently released NHSHC data dashboard.[13] A series of reports will examine the delivery of the programme, prevention opportunities identified and the impact of the NHSHC. The objectives of this first paper are to describe the data extract and to provide an overview of the programme, reporting on: (1) its uptake, process and delivery, (2) the sociodemographic and risk factor profiles of attendees and non-attendees and (3) advice, referrals and statin prescriptions following the check.

## METHODS
### Study setting
Public Health England (PHE) is responsible for national oversight and implementation support of the NHSHC programme. PHE worked with NHS Digital (NHSD) to develop business rules for a data extract of all NHSHC coding activity to allow England wide monitoring of the NHSHC.[14] A Data Extract Advisory Committee (DEAC) was set up to guide use of the data extract. Full details of the scope and composition of the committee are available online.[15]

### Study design
We conducted a retrospective descriptive cross-sectional study of all individuals who were offered an NHSHC, using individual-level participant data. We describe the data extraction before defining the study population. The study design and report conform to the REporting of studies Conducted using Observational Routinely-collected Data (RECORD) recommendations for reporting of observational studies using routinely collected data.[16]

Data were extracted from 6524 (90%) of the 7216 general practices (GPs) participating in the General Practice Data Extraction Service (GPES),[17] after excluding individuals who had opted out of their data being used for purposes other than direct patient care.[18]

The inclusion criteria for the data extract were primary care Read code for any one of the following NHSHC activities: invitation, completion, non-attendance, inappropriate, commenced or declined (prior to 1 April 2018). Full details of the Read codes used for defining NHSHC activity are available in online supplemental table 1.

The data extracted for each individual included sociodemographic characteristics, risk factors for CVD, diagnostic tests requested following the check and interventions including advice and referrals. CVD diagnoses and medication data were also extracted from three out of the four GP clinical information technology system providers, corresponding to 60% of practices. Data extraction for all variables was restricted to time windows around the individual's contact with the NHSHC programme as specified in the business rules for extraction, listed in online supplemental table 2. Data for CVD diagnoses and a broader range of medications will be presented in subsequent papers.

At the time of extraction in 2018, the business rules limited the upper age limit to 75 years for each year. Due to the rolling nature of the programme, this resulted in missing data for the 70–74 age group, most of whom turned 75 during the 5-year cycle. Thus, the maximum age of patients in the extract is 69 for the financial year 2012/2013, compared with 73 in 2016/2017. The final extraction consisted of 12 151 896 patient records with NHSHC activity coding recorded up until 31 March 2018. Data management and data cleaning details are provided in Supplementary Methods and online supplemental table 3.

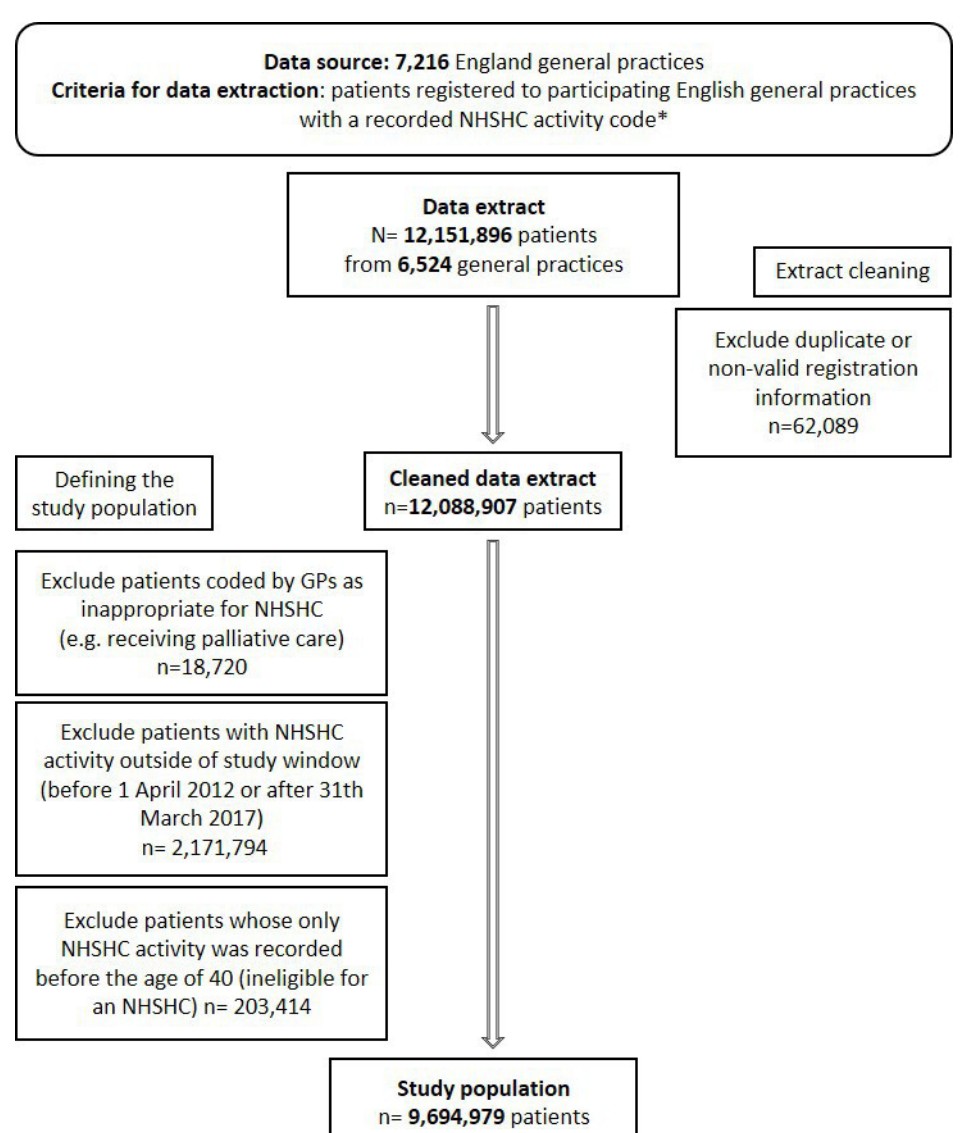

**Figure 1** Study extract and study population flowchart. The study population inclusion dates (1 April 2012 to 31 March 2017) reflect a snapshot of the 5-year rolling programme from April 2012, when all trusts commissioning primary care in England had implemented the programme. *NHSHC activity refers to any interaction that a patient may have had with the NHSHC programme. This includes if a patient was invited to, commenced, completed, declined, did not attend, or was inappropriate for, the NHSHC. More details are provided in online supplemental table 1. GP, general practices; NHSHC, National Health Service Health Check.

## Study population

NHSHCs are offered to individuals aged 40–74 years and without any of the following conditions: hypertension, diabetes mellitus, familial hypercholesterolaemia, coronary heart disease, heart failure, atrial fibrillation, stroke or transient ischaemic attack, peripheral arterial disease, chronic kidney disease and those already on statins or known to have a 10-year CVD risk of ≥20%.[5]

The study population for this analysis was derived from the data extract described above for any NHSHC coded activity. From this group, individuals (1) with NHSHC activity coded outside the study window, (2) aged <40 years at the time of activity and (3) coded by the GP as inappropriate for an NHSHC were then additionally excluded. The final study population thus included only those people offered an NHSHC (invited or completed). Figure 1 presents the study extract and population flowchart.

## Definitions and study variables

Individuals were categorised as either NHSHC attendees if they had a Read code for a completed check within the 5-year period or a non-attendee if they did not. Uptake of the programme was defined as the proportion of the total study population who attended.

An index date was generated from the date of an individual's primary NHSHC activity to identify age and the most relevant risk factor measurements for each patient. Risk factor and clinical measurements were selected for analysis if they occurred on the index date. Otherwise

we took the closest recording within predefined time windows set by the DEAC. Statin prescriptions that occurred on or after the index date among attendees with no data for previous statin prescription were selected. A full list of variables, Read codes used to define variables, time windows and coding algorithms are available in online supplemental table 4.

Further details on study variable definitions and thresholds are provided in Supplemental Methods and online supplemental tables 4–8.

### Data presentation

Statistical tests were not used for comparisons because the amount of missing data between groups varies, preventing meaningful comparisons and the large size of the study population permits the identification of very small differences between groups. Instead, we highlighted the size of differences between groups and interpreted it in relation to the missing data. Where appropriate, we presented data for attendees and non-attendees. Data for uptake, invitation type and third-party provider are presented by financial year to describe changes over time. Data on uptake are also presented by local authority for geographical comparisons. To minimise bias, we include missing data details in all tables and figures.

### Patient and public involvement

PHE developed an information notice for patients, including an easy read version, explaining how their personal data would be used and the purpose of the research project. Membership of the DEAC overseeing the use of the NHS Health Check dataset, including the development of this study, its design and outcomes, includes a patient representative. Study results will not be disseminated to individuals whose data are used but the collective analysis presented here will be shared publicly once published.

### Ethical approval

A Direction from the Secretary of State for Health and Social Care instructed NHS Digital with the legal requirement to carry out the NHSHC data extract.[19] This study was subject to an internal review by the Research Support and Governance Office in PHE to ensure that it was fully compliant with the UK Policy Framework for Health and Social Care Research (2017) and with all other current regulatory requirements.

### RESULTS
### NHSHC uptake
#### Overall uptake by year

Between 1 April 2012 and 31 March 2017, 9 694 979 individuals aged 40–74 years were offered an NHSHC in England. Of these 5 102 758 (52.6%) completed a check. Uptake by financial year is presented in table 1. Uptake remained >50% throughout the 5 years of programme delivery. The number of individuals offered an NHSHC

**Table 1** Attendance to an NHS Health Check by financial year among individuals aged 40–74 years in England between April 2012 and March 2017 (N=9 694 979)

| Financial year | Individuals offered an NHS Health Check | Individuals attending an NHS Health Check | Uptake of offers rate % |
|---|---|---|---|
| 2012/2013 | 1 469 031 | 742 935 | 50.6 |
| 2013/2014 | 1 796 483 | 962 831 | 53.6 |
| 2014/2015 | 2 162 454 | 1 135 746 | 52.5 |
| 2015/2016 | 2 154 129 | 1 142 151 | 53.0 |
| 2016/2017 | 2 112 882 | 1 119 095 | 53.0 |
| Total | 9 694 979 | 5 102 758 | 52.6 |

NHS, National Health Service.

increased from just under 1.5 million in 2012/2013 to 1.8 million the year after, plateauing at approximately 2.1 million each year after that (table 1).

#### Geographical variation in uptake of offers

Across England, uptake rates varied by region, as presented in figure 2A. The highest uptake of offers over the 5-year cycle was 84.7% and the lowest 25.1% by region. Data for uptake by upper tier local authority are available in online supplemental table 9. Variation in uptake in London is shown in figure 2B. Central and north London local authorities had higher rates of uptake, with lower rates in the south east.

### Process and delivery
#### Invitation frequency

Of the 9 694 979 individuals in the study population with codes for NHSHC activity, 7 970 396 (82.2%) had a record of at least one NHSHC invitation (see online supplemental table 10). Table 1 presents the number of recorded invitations for attendees and non-attendees (recording by each financial year is available in online supplemental table 11).

Among the 5 102 758 attendees, almost a third (32.8%) had no invitation code recorded but still had a completed NHSHC recorded. The remaining two-thirds (3 429 914) had an invitation recorded, with 50.5% having one invitation and 16.7% having two or more. Among these attendees coded as invited, 590 869 (17.2%) received an invitation on the same date as the NHSHC and were thus assumed to be opportunistic rather than planned. Among those with an invitation in advance of the NHSHC (82.8%; n=2 839 045), the median number of days between recording of their first invitation and a completed NHSHC was 42 (IQR 21, 90) days.

Among non-attendees, 98.9% had a formal invitation record, with a quarter (25.5%) having two or more invitations. The remaining 1.1% of non-attendees had Read codes for declining or not attending a check (see online supplemental table 1).

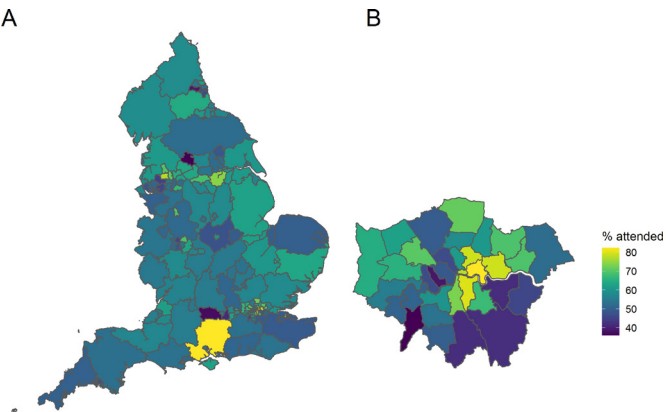

**Figure 2** Variation in NHSHC uptake across (A) England and (B) London. Uptake rates shown as % of people taking up an offer of a check, between 2012/3 and 2016/17, by upper tier local authority of the individuals' usual residence. NHSHC, National Health Service Health Check.

### Invitation type

Among both attendees and non-attendees, the most common invitation type was a letter, however, other forms of invitations, including text messaging, increased with each year of the programme. The online supplemental figure 1 presents the type of invitation by financial year among attendees and non-attendees.

### Delivery

Among all attendees within the 5-year time frame, 3.0% had a clinical code to indicate that their NHSHC was completed by a third party. This increased gradually from 1.2% in the first year to 4.1% in the final year.

### Characteristics of invitees
#### Sociodemographic characteristics

Table 2 presents the sociodemographic characteristics of the study population and the characteristics of the general population according to Office for National Statistics modelled estimates. The population offered an NHSHC was representative of the general population of people aged 40–74 years in terms of sex and deprivation index although they were younger relative to the age distribution of the general population (age <55: 62.2% vs 49.7%). Those who were offered an NHSHC also closely resembled the ethnic makeup of the general population for most ethnicities, except for people self-reporting as white or black Caribbean who appeared underrepresented, although 16.7% of data for ethnicity were missing.

Attendees differed from non-attendees. More attendees were women (54.7%) compared with non-attendees (47.5%; general population 50.9%). There were also notable differences by age. Most attendees were <55 years as they constituted the largest group of eligible people, but individuals ≥55 years had higher rates of attendance after invitation. For ethnic group comparisons, a large proportion of missing data for non-attendees (27.8%) compared with attendees (6.8%) limits interpretation, but where data were available and compared with the

general population, ethnic minority groups appeared to be better represented among attendees than non-attendees (table 2).

Deprivation indices indicate few differences between attendees and non-attendees, except at the extreme ends of the index of multiple deprivation spectrum, where there were slightly more attendees from the most affluent areas (Decile 10: 11.0% vs 10.0%) and slightly fewer attendees from the most deprived areas (Decile 1: 8.2% vs 9.4%). Finally, although the numbers were small, there was no evidence to indicate that people with severe mental illness, physical or cognitive disability were underrepresented among attendees (table 2).

### Risk factors

Overall, completeness of data for common risk factors measurements including systolic blood pressure (95.8%), smoking (95.7%), Body Mass Index (BMI) (96.3%) and total cholesterol (93.6%) was high in attendees, in contrast to recording of physical activity (64.5%), blood glucose (18.2%), Haemoglobin A1C (HbA1C) (36.6%) and alcohol (38.3%). A CVD risk score was formally documented for 79.7% of attendees (figure 3, online supplemental table 12). Family history data were only recorded where a positive finding was present, making it difficult to estimate how much data were missing or were assessed and were negative. Completeness of most, but not all risk factors, was lower among non-attendees, with the exception of diabetes risk measurements that were similarly low in both groups.

Figure 4 shows the proportion of all individuals identified as having each CVD risk factor among attendees and non-attendees and with respect to missingness of data. Among attendees, where missingness was low, we identified 24.5% with hypertension, while 23.8% were obese and 16% were current smokers. Where a 10-year CVD risk score was documented in the primary care record (79.7% of attendees), just over a quarter (25.9%) were identified as high risk, with a score of ≥10%.

### Interventions
#### Advice, information and referrals

Advice, information and referral for an intervention following an NHSHC were recorded almost 6 million times for all attendees and more than 2.5 million times for individuals with elevated CVD risk factors (table 3). Among all attendees, 16.0% were coded to have received general lifestyle and behavioural advice, just over a fifth were given formal advice on diet and almost a third on physical activity. Among those whose alcohol use puts them above low risk, more than a third were directed to alcohol treatment services. Almost half of all current smokers were directed to smoking cessation services and 19.6% of those who had BMI ≥30 were directed to weight loss and obesity services.

### Statin prescriptions

Information on a new statin prescription, occurring on or after NHSHC completion, was available for 60.4% of all attendees

**Table 2** Sociodemographic characteristics of NHSHC invitees April 2012–March 2017 compared with ONS estimated English population aged 40–74 at mid-2015

| Sociodemographic characteristic | ONS mid-2015 England resident population (aged 40–74 years) n (%) | NHSHC invitees n (%) | Attendees n (%) | Non-attendees n (%) |
|---|---|---|---|---|
| Sex | | | | |
| Male | 11 200 690 (49.1) | 4 724 015 (48.7) | 2 311 604 (45.3) | 2 412 411 (52.5) |
| Female | 11 604 922 (50.9) | 4 970 906 (51.3) | 2 791 130 (54.7) | 2 179 776 (47.5) |
| Unknown | – | 58 (0.0) | 24 (0.0) | 34 (0.0) |
| Age group (years) | | | | |
| 40–44 | 3 636 454 (15.9) | 2 208 213 (22.8) | 984 908 (19.3) | 1 223 305 (26.6) |
| 45–49 | 3 889 360 (17.1) | 1 986 966 (20.5) | 966 356 (18.9) | 1 020 610 (22.2) |
| 50–54 | 3 811 000 (16.7) | 1 833 267 (18.9) | 958 263 (18.8) | 875 004 (19.1) |
| 55–59 | 3 278 322 (14.4) | 1 414 091 (14.6) | 783 740 (15.4) | 630 351 (13.7) |
| 60–64 | 2 904 721 (12.7) | 1 105 914 (11.4) | 669 503 (13.1) | 436 411 (9.5) |
| 65–69 | 3 017 135 (13.2) | 910 089 (9.4) | 585 653 (11.5) | 324 436 (7.1) |
| 70–74 | 2 268 620 (9.9) | 236 439 (2.4) | 154 335 (3.0) | 82 104 (1.8) |
| Ethnic group | | | | |
| White | 20 383 677 (89.4) | 6 946 824 (71.7) | 4 067 864 (79.7) | 2 878 960 (62.7) |
| Indian | 524 313 (2.3) | 202 004 (2.1) | 136 598 (2.7) | 65 406 (1.4) |
| Pakistani | 291 546 (1.3) | 137 222 (1.4) | 89 970 (1.8) | 47 252 (1) |
| Bangladeshi | 101 926 (0.4) | 46 802 (0.5) | 34 863 (0.7) | 11 939 (0.3) |
| Black African | 314 107 (1.4) | 147 462 (1.5) | 94 539 (1.9) | 52 923 (1.2) |
| Black Caribbean | 271 649 (1.2) | 79 987 (0.8) | 53 621 (1.1) | 26 366 (0.6) |
| Chinese | 121 129 (0.5) | 44 730 (0.5) | 27 360 (0.5) | 17 370 (0.4) |
| Other Asian | 302 667 (1.3) | 125 853 (1.3) | 79 354 (1.6) | 46 499 (1) |
| Other group | 494 599 (2.2) | 239 024 (2.5) | 142 621 (2.8) | 96 403 (2.1) |
| Not stated | | 104 136 (1.1) | 31 319 (0.6) | 72 817 (1.6) |
| Missing | | 1 620 935 (16.7) | 344 649 (6.8) | 1 276 286 (27.8) |
| Deprivation index (IMD decile) | | | | |
| Most deprived | 1 914 356 (8.4) | 853 547 (8.8) | 420 547 (8.2) | 433 000 (9.4) |
| 2 | 1 999 183 (8.8) | 896 809 (9.3) | 472 647 (9.3) | 424 162 (9.2) |
| 3 | 2 083 743 (9.1) | 904 131 (9.3) | 477 140 (9.4) | 426 991 (9.3) |
| 4 | 2 202 902 (9.7) | 921 244 (9.5) | 477 516 (9.4) | 443 728 (9.7) |
| 5 | 2 304 663 (10.1) | 974 023 (10) | 509 715 (10.0) | 464 308 (10.1) |
| 6 | 2 402 719 (10.5) | 991 135 (10.2) | 517 381 (10.1) | 473 754 (10.3) |
| 7 | 2 443 073 (10.7) | 1 044 505 (10.8) | 547 909 (10.7) | 496 596 (10.8) |
| 8 | 2 458 761 (10.8) | 1 034 751 (10.7) | 547 016 (10.7) | 487 735 (10.6) |
| 9 | 2 491 679 (10.9) | 1 045 098 (10.8) | 565 872 (11.1) | 479 226 (10.4) |
| Least deprived | 2 504 533 (11.0) | 1 022 539 (10.5) | 563 798 (11.0) | 458 741 (10.0) |
| Missing | | 7197 (0.1) | 3217 (0.1) | 3980 (0.1) |
| Patient characteristics | | | | |
| Deaf | n/a | 321 (0.0) | 171 (0.0) | 150 (0.0) |
| Blind | n/a | 13 405 (0.1) | 7224 (0.1) | 6181 (0.1) |
| Severe mental illness | n/a | 111 878 (1.2) | 59 351 (1.2) | 52 527 (1.1) |
| Learning disability | n/a | 39 612 (0.4) | 21 535 (0.4) | 18 077 (0.4) |
| Dementia | n/a | 7521 (0.1) | 3060 (0.1) | 4461 (0.1) |

Continued

**Table 2** Continued

| Sociodemographic characteristic | ONS mid-2015 England resident population (aged 40–74 years) n (%) | NHSHC invitees n (%) | Attendees n (%) | Non-attendees n (%) |
|---|---|---|---|---|
| Rheumatoid arthritis | n/a | 74 281 (0.8) | 38 104 (0.7) | 36 177 (0.8) |
| Total | 22 805 612 | 9 694 979 | 5 102 758 | 4 592 221 |

IMD, index of multiple deprivation; NHSHC, National Health Service Health Check; ONS, Office for National Statistics.

(n=3 079 705, see the Methods section). Overall, a statin was prescribed for 8.2% of these attendees. Stratifying this group by CVD risk revealed that a statin was prescribed in 20.3% of those with a 10-year CVD risk score ≥10% and in 39.1% of those with a CVD risk score of ≥20%. Among the 1 910 919 individuals with a CVD risk score <10%, 3.3% received a new statin prescription, while in the remaining 504 374 with no CVD risk score recorded, 11.0% were prescribed a statin (see online supplemental table 13).

Assuming similar rates of statin prescription nationally, we estimate that of the 5 102 758 attendees in this study, up to 418 000 may have received a new statin prescription, with over half of these (n~213 000) prescribed to those identified at the NHSHC visit as being at >10% risk of CVD events.

## DISCUSSION

In the largest nationwide study of the NHS Health Check programme, using primary care data, we find that the checks have been offered to over 9.5 million people during a 5-year cycle up to 2017, with 52% of people taking up the offer. While we noted geographical variation in uptake rates and an age and sex bias for attendance, we found little evidence of inequality in who was offered or who received an NHSHC by ethnicity or deprivation indices. Where an NHSHC was delivered, risk factors were identified at a similar rate to population estimates, with advice and referrals offered over 2.5 million times to those with risk factors, along with 20% of those at highest risk receiving a new statin prescription as per guidelines. These insights into the evolving process and delivery of the NHSHC programme will support efforts to further enhance the value of the programme, especially for improving uptake rates, targeting those at greatest risk

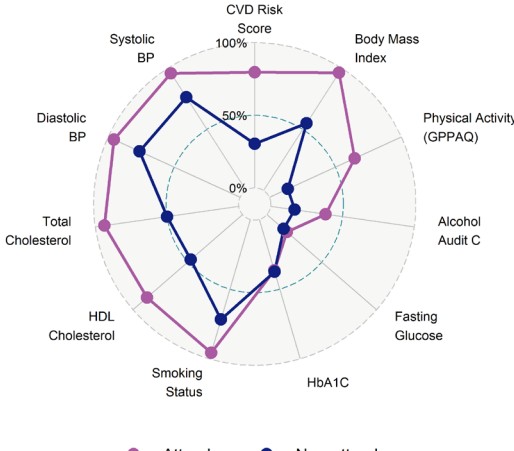

**Figure 3** Completion of risk factor measurements for attendees and non-attendees (2012/13–2016/17). Proportion of available and missing data for each risk factor related measurements are shown here. Note these are available measurements within the time frame of the data extract (see Supplemental Methods). Family history not shown as coded only as yes with unknown negative/missing data. See also online supplemental table 12 for the completeness values. AUDIT-C, Alcohol Use Disorders Identification Test-Consumption; BP, blood pressure; CVD, cardiovascular disease; HbA1C, haemoglobin A1c; HDL, high-density lipoproteins; GPPAQ, General Practice Physical Activity Questionnaire.

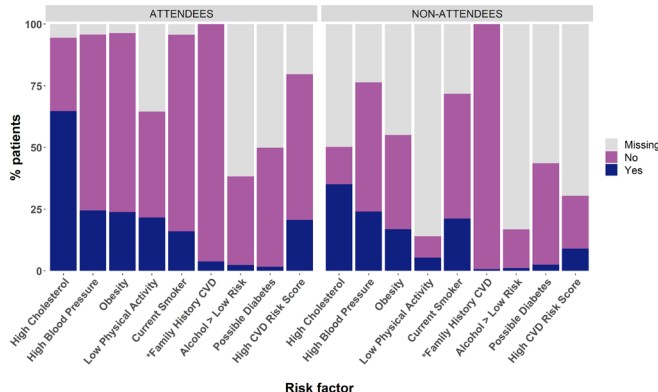

**Figure 4** Proportion of attendees and non-attendees with common CVD risk factors. Definitions as per online supplemental table 6) and include: high cholesterol=total cholesterol >5 mmol/L or cholesterol ratio >4; high blood pressure=systolic ≥140 or diastolic pressure ≥90 mm Hg; obesity=body mass index≥30 kg/m$^2$; alcohol>low risk=Alcohol Use Disorders Identification Test-Consumption (AUDIT C) score ≥8; low physical activity=General Practice Physical Activity Questionnaire (GPPAQ) moderate inactive or inactive; possible diabetes= haemoglobin A1C (HbA1C) ≥48 mmol/mol or Fasting Blood Glucose (FBG) >7 mmol/L; current smoker=current smoking; high CVD risk score=10-year CVD risk score ≥10%. *Family history is predominantly only recorded if present so accurate information on its absence is unavailable. See also online supplemental table 6 for more detailed information. CVD, cardiovascular disease.

**Table 3** Number and proportion of attendees that were coded as received advice, information or a referral following their NHSHC among all attendees and attendees with CVD risk factors

| Intervention type | All attendees n (%) | Attendees with the CVD risk factor above threshold for intervention n (%) |
|---|---|---|
| Alcohol consumption | 792 761 (15.5) | 46 611 (38.4) |
| Diet | 1 189 986 (23.3) | 766 521 (25.1) |
| Physical activity | 1 501 103 (29.4) | 434 326 (39.3) |
| General lifestyle/ behaviours | 814 611 (16.0) | 211 571 (20.1) |
| Smoking cessation | 865 913 (17) | 467 119 (57.3) |
| Weight loss and obesity | 821 414 (16.1) | 599 380 (19.6) |
| Diabetes prevention programme | 4551 (0.1) | 3348 (0.9) |
| Total | 2 501 565 (49.0) | 565 047 (53.7) |

Thresholds defined in online supplemental table 8.
CVD, cardiovascular disease; NHSHC, National Health Service Health Check.

and maximising the use of available NCD and CVD risk reduction interventions.

Our key finding of a 52% uptake rate is slightly higher than previous studies, reporting around 48%.[10] This may be due to the larger, more nationally representative and contemporary data to which we had access, supported by the finding that uptake rates have steadily increased since 2012. Furthermore, we also found wide geographical variation, across the country and in London, possibly due to differing coding practices or invitation methods, which could skew findings from smaller studies or explain discordance with other reports of NHSHC activity.[20] However, an important difference that precludes direct comparison with other studies reporting on NHSHC reach is that our study was restricted to people who had an NHSHC code in their GP records, indicating either an invitation or completion of a check. As such we were unable to quantify coverage of the programme, that is, how many eligible people were offered a check. Estimates from PHE, based on Office for National Statistics data minus the estimated number of people on existing disease registers suggests an eligible population of ~15.5 million.[20] Using this number and based on 5.1 million having had a check, we estimate that a further 6.5 million in the same 5 year cycle would need to complete an NHSHC to achieve the original programme aspiration of 75% coverage.[4 8]

Some NHSHC providers have raised concerns that the programme may paradoxically increase health inequality by only attracting the worried well with more affluent and white people.[21] Reassuringly the data do not show gross differences in the offering or uptake of the programme. First, those who were offered an NHSHC closely resemble the population of England, as measured through census data, with no differences by sex, ethnicity or deprivation indices. They were slightly younger overall, but this is likely because eligibility for an NHSHC falls with comorbidities which are frequently age related.[5] Second, although missing data on ethnicity limit definitive conclusions, ethnic minorities such as those from South Asian backgrounds were equally if not more represented as reported by others.[22 23] Furthermore, although there were small differences at the extremes of deprivation deciles, overall, there was no gross bias towards greater attendance by increasing affluence and previous mixed findings are likely due to regional variation,[22–24] while the similar uptake rates in those with physical disability or serious mental illness also indicate that the programme is equitably delivered. There was however a notable bias towards more women and older people attending for an NHSHC compared with non-attendees, a finding also observed by others.[10 11 22 23]

Of note, despite older people being more likely to attend than not attend after having an offer of an NHSHC, proportionally 57% of all attendees were <55 years, which is higher than reports from other national evaluations of the programme.[11] This could be because our data were limited for the age 70–74 group or that more older people are excluded having been identified with comorbidities earlier in the programme cycle when these other studies reported. However, it may also indicate that younger people are motivated to understand their CVD risk and engage with care providers to address their longer term and lifetime risk, a finding we previously observed with the use of digital risk assessment tool.[25] The potential benefits of this earlier engagement with CVD risk will need to be evaluated over the longer term.

An important benefit of the NHSHC programme has been improvements in risk factor and behaviour data recording, which can guide patient interventions and inform regional resource priorities. For core data items such as smoking status, data completeness was as high as 96%, while for alcohol and physical activity (measures that are legally required as part of the NHSHC but not needed to calculate a person's 10-year CVD risk) was close to 65%. This contrasts with the high degree of missing data among non-attendees for most risk factors. The exception being blood glucose and HbA1C measurements which were similarly complete at low levels for both non-attendees and attendees. This may be because these tests are only performed in attendees at high diabetes risk, combined with parallel current or historical efforts to establish and maintain a diabetes disease register outside of the NHSHC. Where risk factors were recorded, they reveal that prevalence in attendees is close to those in the wider UK population.[3 26] A 10-year risk score was documented in 79.7% of all attendees. We anticipate that in the remaining ~20%, practitioners may have estimated

the score using an online or other tool not integrated into the clinical system, which may have meant that the score was discussed but not recorded, although it is possible some may not have calculated it at all. Overall, where a score was recorded over a quarter of all attendees were calculated to have a 10 year CVD risk score of ≥10%, the current threshold set by the National Institute of Clinical Excellence (NICE) to consider preventative interventions such as statin prescription.[27] Indeed, we found that 20% of this population was newly prescribed a statin following the NHSHC. This figure was even higher at nearly 40% for those with a 10-year CVD risk score of ≥20%, an older NICE threshold for statin prescription. This is an encouraging finding, being higher than in earlier studies and approaching the national ambition of 45% for statin use in this very high risk group.[11 28] Our data also suggest that the NHSHC encounter prompted relevant non-statin interventions with over 2.5 million people with risk factors being coded as having received advice, information or referrals. We note however that these figures may be an underestimate being entirely dependent on coding practices and availability of services by region. For example, the low referral rates for the diabetes prevention programme) are partly explained by the programme launching relatively recently in 2016 and also due to variation in its availability across England and the poor recording of referrals to the programme in the primary care record as reported by others.[29]

## Limitations

Despite being the largest national evaluation of the NHSHC programme, our study has some important limitations. First, our data were restricted to people with an NHSHC activity code, and thus we were unable to quantify the full eligible population to determine coverage and the gap in programme reach. Although this is an aspiration for future analyses, it will require access to GP records for much of the population, raising important data governance and handling challenges. Second, we had substantial missing data, especially for the non-attendees, limiting our ability to make robust conclusions about differences in characteristics and risk between these groups. Also, our data extract did not include information on 10% of practices in GPES, which could have introduced a degree of bias in our estimates if the reasons for missing data were not random and related to participation in the NHSHC programme. Third, important information on those >70 years was limited due to a business rule that led to loss of older people once they turned 75 for each year of the data extract. However, the proportionally smaller number of older people eligible for an NHSHC means our results are unlikely to have been impacted significantly. Fourth, prescription data were only available from 60% of practices. The estimate for statin prescriptions derived from the available data however is likely to be representative. Finally, we used a Read code to identify whether an NHSHC took place. This, of course, does not provide any indication as to the extent or quality of the conversations around risk or the suitability of information given, on which the full impact and value of an NHSHC are likely to depend.

## Clinical implications

This analysis provides a national-level overview of the NHSHC programme, against which local authorities and healthcare providers can benchmark local achievements. Used with the NHSD dashboard, this will enable local CVD risk strategies to be developed, to increase the invitation of eligible individuals not yet invited for an NHSHC as well as targeting those who still do not attend even after invitation.[13] Importantly, we show that a national prevention programme to tackle NCDs is possible and population health can be targeted through routine healthcare. It represents a systematic approach to switching the conversation from illness to preventing disease and appears to have good engagement from the public so far. From the data, we observe that in England, there remains a major challenge for reducing risk factors that impact multiple long-term chronic conditions. The programme appears to have been successful at promoting advice and guideline-based interventions. Although assessing the efficacy of these interventions on individual-level behaviour change is challenging, further analysis of this large dataset will explore the impact on available metrics such as diagnosis rates and clinical outcomes.

## CONCLUSION

In this large-scale analysis of the NHSHC programme using national primary care data, we found that in recent years, over half of all people offered a check have completed one. Although there was substantial variation between local authorities in uptake rates, we found little or no evidence of inequity in invitation processes or uptake. Furthermore, the programme has identified a high burden of risk among attendees, with correspondingly encouraging levels of guideline-driven advice, referrals and statin prescriptions for the primary prevention of CVD. However, to achieve fully the anticipated benefits of the NHSHC programme, we highlight a need for continued efforts to invite more of the eligible population for an NHSHC, reduce geographical variation in uptake of offers, prioritise those who are not attending and to maximise the use of evidence-based interventions to support risk reduction. Subsequent research should provide more insight into how different delivery models influence outcomes.

**Author affiliations**
[1]Institute of Cardiovascular Science, University College London, London, UK
[2]Public Health England, London, UK
[3]NHS Digital, Leeds, UK
[4]Department of Cardiovascular Sciences, University of Leicester, Leicester, UK
[5]NIHR Leicester Biomedical Research Centre, Glenfield Hospital, Leicester, UK
[6]Centre for Primary Care and Public Health, Queen Mary University of London, London, UK
[7]UCL Partners, London, UK

**Acknowledgements**  We would like to thank colleagues from PHE and NHS Digital who supported this work. We would also like to thank the patient and public representatives involved with this work, for their input.

**Contributors**  All authors contributed to conception of the study, study design, overall analysis plan and critically reviewed the final manuscript. Specifically in addition, RP, SB and KT contributed to the statistical analysis plan, review of results and drafted and revised the final paper; SB, CL, EC, TE and RW obtained and analysed all data and contributed to drafting of the final manuscript; SC, JF and DR supported data extraction for the analysis and review of the final manuscript; MN, NJS, JR critically reviewed and edited the paper; MK, JD, JW conceived the study; contributed to the analysis plan and critically reviewed the final manuscript.

**Funding**  RP (FS/14/76/30933) and JD (BHF chair) were funded by the BHF. Data extraction and analysis were funded by PHE.

**Map disclaimer**  The depiction of boundaries on the map(s) in this article does not imply the expression of any opinion whatsoever on the part of BMJ (or any member of its group) concerning the legal status of any country, territory, jurisdiction or area or of its authorities. The map(s) are provided without any warranty of any kind, either express or implied.

**Competing interests**  None declared.

**Patient consent for publication**  Not required.

**Ethics approval**  The review also covered all ethical considerations. No ethical issues were identified and thus review by an ethics committee was not required (Personal communication between Katherine Thomson & PHE Research Support Governance Office, 2019).

**Provenance and peer review**  Not commissioned; externally peer reviewed.

**Data availability statement**  Data are available upon reasonable request. All data relevant to the study are included in the article or uploaded as supplemental information. The legal basis for the data extract was a Secretary of State for Health and Social Care Direction. With DEAC approval PHE and NHS Digital have set up a process for dealing with information requests relating to the pseudonymised primary care data used in this paper. The purpose for using this data must be for the scope of work relating to the evaluation of the NHS Health Check in line with the requirements of the Direction.

**ORCID iD**
Riyaz Patel http://orcid.org/0000-0003-4603-2393

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
