## [Reviewer comments · BMJ Open]

ARTICLE DETAILS

TITLE (PROVISIONAL)	An evaluation of the uptake and delivery of the NHS Health Check Programme in England, using primary care data from 9.5 million people: A cross-sectional study
AUTHORS	Patel, Riyaz; Barnard, Sharmani; Thompson, Katherine; Lagord, Catherine; Clegg, Emma; Worrall, Robert; Evans, Tim; Carter, Slade; Flowers, Julian; Roberts, Dave; Nuttall, Michaela; Samani, Nilesh; Robson, John; Kearney, Matt; Deanfield, John; Waterall, Jamie

VERSION 1 - REVIEW

REVIEWER	Janet Krska University of Kent, UK
REVIEW RETURNED	04-Aug-2020

GENERAL COMMENTS	This is an important paper covering a very extensive set of data from a national CVD prevention programme, which in itself is of international importance. The paper is well written and clear. It focuses on two key aspects of such a programme - uptake and delivery. It does not attempt to look at clinical outcomes, but does report statin prescription. Conclusion is drawn from the findings. I drafted some specific comments below which may help improve the clarity of the paper in a few places and also add to discussion. The main issue I have is with the bulleted section on Strengths and limitations, which does not cover these adequately. Limitations are included in the main text, however this section needs to be re-drafted. Objectives of the work are not clearly defined as such. Although the purpose of this paper is set out, the text could be re-drafted to make the objectives of the present paper clearer. Abstract Results should make it clear that only approximately 80% of attendees had a CVD risk score documented and that it was 20.6% of these who had a 10-year CVD risk $\geq 10\%$. Also add that the 20.3% of those at increased CVD risk who were prescribed a statin was derived from a subsample which represented only 60.4% of the total. A further key addition could be to add the actual number who received an NHSHC during the 5 year period i.e. 5,102,758. Limitations are listed in main paper, but not in Strengths and limitations section, which requires re-drafting in line with journal requirements. One of the points in this section states that the paper
---

reports...“subsequent use of risk modifying interventions”. It would be clearer to state that the paper reports advice and referrals given.

Methods page 5

“Data was extracted from 6,524 (90%) of the 7,216 General Practices participating in the General Practice Data Extraction Service (GPES)” – how many practices provide NHSHC and are not in the GPES? Should this be included and how does it affect the results?

Text states: “The data extracted for each individual included socio-demographic characteristics, risk factors for cardiovascular disease, diagnostic tests, and interventions including advice and referrals.”

Not entirely clear what is meant by diagnostic tests here – it doesn't seem to be tests requested for/recorded at the NHSHC. Would it be more accurate to say the results of diagnostic tests and also that these include previous tests or those carried out/recorded at the NHSHC? While what was extracted is clear from the Supplementary table, some clarification here may make reading easier without having to refer to this table.

Text states that “CVD diagnoses and medication data were also extracted from three out of the four GP clinical IT systems providers, corresponding to 60% of practices.” CVD diagnoses are not reported in this paper. The only medication data reported is statins. So again this sentence is slightly misleading to the reader and could perhaps be changed to relate to the results presented here.

While it is true to say that “individuals with current statin prescriptions would not be eligible for an invitation to the NHSHC” it is entirely possible that despite this, some may still have been invited. Data extraction does not seem to take account of this possibility, so is there an assumption that any statin prescription is newly initiated and is this a valid assumption?

Results

Page 11 It is not entirely correct to that that “Completeness of all risk factors was lower among non-attendees” as more non-attenders had HbA1c recorded. There were similar proportions with records of fasting glucose and recording of smoking and blood pressure are also pretty high in non-attenders. Suggest modify Discussion slightly to reflect this.

Page 12 “Advice, information and referral recorded almost six million times for all attendees, and more than 2.5 million times for individuals with elevated CVD risk factors.” What proportion of those with high risk score were given advice compared to those with <10% risk score? What was the actual number/proportion of attendees given any advice and the number with one, two, three pieces of advice/referral? (Potentially useful as the more people are asked to change, the less they are likely to do so.)

Table 3 Referrals to DPP are given, but isn't it the case that this had not been launched at the start of this data extraction period? No comment in Discussion on the small number of such referrals.

The proportion in the general practices from which prescribing data was extracted who had a 10% CVD risk score or above was 21.6%, slightly higher than for full sample (20.4%). This is not stated.

Key for Figs 1 and 2 (especially Fig 1) could be more useful – it's quite hard to see what % the different colours represent. Why not include all reported levels in both?

Is Fig 3 format the best way of showing these data?

	Fig 4 may be better if each risk factor was presented for both attendees and non-attenders together, enabling easier comparisons. Perhaps consider horizontal bar chart. Discussion A CVD risk score was formally recorded for 79.7% of attendees. In theory it should be 100%. No comment is made on whether this is considered a good or a poor result. “The programme appears to have been successful at promoting advice and guideline-based interventions.” While it is true to say that the extent or quality of the conversations around risk or the suitability of information given is not known, it is also a limitation that no data are reported on whether advice was followed, referrals accepted etc and actual clinical outcomes or even proxy outcomes (reduced BP, cholesterol, weight). Such outcomes are not part of this work (and may not be recorded). While this seems obvious, it may be useful to just add this point to the text.
--	--

REVIEWER	Lars Bruun Larsen University of Southern Denmark, Denmark
REVIEW RETURNED	18-Aug-2020

GENERAL COMMENTS	Thank you for the opportunity to review this interesting manuscript. I do not have any substantial comments. The background and methods sections are clear and well-written, and the authors present a balanced discussion of the results and the limitations due to e.g. missing values. The results are very interesting, especially given the large dataset and availability of information.
---

VERSION 1 – AUTHOR RESPONSE

Reviewer: 1

This is an important paper covering a very extensive set of data from a national CVD prevention programme, which in itself is of international importance. The paper is well written and clear. It focuses on two key aspects of such a programme - uptake and delivery. It does not attempt to look at clinical outcomes but does report statin prescription. Conclusion is drawn from the findings.

We thank the reviewer for these supportive comments

The main issue I have is with the bulleted section on Strengths and limitations, which does not cover these adequately. Limitations are included in the main text, however this section needs to be re-drafted.

Thank you, we have edited the section as advised. Please see the response to Editorial team comments above.

Objectives of the work are not clearly defined as such. Although the purpose of this paper is set out, the text could be re-drafted to make the objectives of the present paper clearer.

We have now amended the text at the end of the introduction to make the objectives of the study clearer, on page 5, paragraph 1

“The objectives of this first paper are to describe the data extract and to provide an overview of the programme, reporting on: (i) its uptake, process and delivery, (ii) the sociodemographic and risk factor profiles of attendees and non-attendees and (iii) advice, referrals and statin prescriptions following the check. “

Abstract Results should make it clear that only approximately 80% of attendees had a CVD risk score documented and that it was 20.6% of these who had a 10-year CVD risk $\geq 10\%$. Also add that the 20.3% of those at increased CVD risk who were prescribed a statin was derived from a subsample which represented only 60.4% of the total. A further key addition could be to add the actual number who received an NHSHC during the 5 year period i.e. 5,102,758.

We thank the reviewer for highlighting this. The 20.6% is the proportion of all attendees who had a CVD risk score of $\geq 10\%$ in the total attendee population not just those in whom it was specifically recorded. We have now corrected this to report instead the proportion among those in whom a CVD risk value was documented in the primary care record, which is higher at 25.9%. Likewise, among the 60.4% with statin data, an almost identical proportion had a CVD risk score recorded (25.8%) which was $\geq 10\%$ and of these, 20.3% were prescribed a statin. We have therefore amended the sentence in the abstract. The specific details and numbers are given in more detail in the results section.

“Among attendees risk factor prevalence reflected population survey estimates for England. Where a CVD risk score was documented, 25.9% had a 10-year CVD risk $\geq 10\%$, of which 20.3% were prescribed a statin. Advice, information and referrals were coded as delivered to over 2.5 million individuals identified to have risk factors.”

In the results section we have also added a sentence to clarify this on page 11, line 15:

“Where a 10-year CVD risk score was documented in the primary care record (79.7% of attendees), just over a quarter (25.9%) were identified as high risk with a score of $\geq 10\%$.”

We have also added the actual number of individuals receiving an NHSHC over the 5-year period in the abstract.

“During the 5-year cycle, 9,694,979 individuals were offered an NHSHC and 5,102,558 (52.6%) took up the offer.”

Limitations are listed in main paper, but not in Strengths and limitations section, which requires re-drafting in line with journal requirements. One of the points in this section states that the paper reports...“subsequent use of risk modifying interventions”. It would be clearer to state that the paper reports advice and referrals given.

We have amended the strengths and limitations section as described above.

We have also edited the point referring to use of risk modifying interventions to state:

“This first data analysis reports on elements relating to uptake, implementation, process and delivery of NHSHCs, the sociodemographic and risk factor profile of both those who did and did not attend a check and rates of advice, referrals and statin prescriptions delivered as part of the check”

Methods page 5

“Data was extracted from 6,524 (90%) of the 7,216 General Practices participating in the General Practice Data Extraction Service (GPES)” – how many practices provide NHSHC and are not in the GPES? Should this be included and how does it affect the results?

We have clarified with NHS digital, that the GPES has comprehensive coverage of registered GP practices in England. The number of practices operating varies each year, so it is not possible to specify how many practices were not included in GPES at the time of the extract, but the number is likely to be very small given the total number of practices in England over the last few years has ranged from 6800-7400. It is unlikely that many practices who deliver NHSHCs are not included in GPES and given the size of the dataset, this is unlikely to have meaningful impact on the results. However, of the 10% of GP practices not included in the data extract from GPES (data not available or provided by GP practices, through choice or technical issues), we cannot tell how many collected NHSHC data or what their data may have looked like. It is therefore possible that there could be some bias introduced here. However, with 90% of practices providing data across the country, we do not anticipate any significant impact on our findings. Nonetheless, we have mentioned this in the limitations section.

“Also, our data extract did not include information on 10% of practices in GPES, which could have introduced a degree of bias in our estimates if the reasons for missing data were not random and related to participation in the NHSHC programme.”

Text states: “The data extracted for each individual included socio-demographic characteristics, risk factors for cardiovascular disease, diagnostic tests, and interventions including advice and referrals.” Not entirely clear what is meant by diagnostic tests here – it doesn’t seem to be tests requested for/recorded at the NHSHC. Would it be more accurate to say the results of diagnostic tests and also that these include previous tests or those carried out/recorded at the NHSHC? While what was extracted is clear from the Supplementary table, some clarification here may make reading easier without having to refer to this table.

Text states that “CVD diagnoses and medication data were also extracted from three out of the four GP clinical IT systems providers, corresponding to 60% of practices.” CVD diagnoses are not reported in this paper. The only medication data reported is statins. So again this sentence is slightly misleading to the reader and could perhaps be changed to relate to the results presented here.

Thank you for highlighting these important points. We have made this clearer by amending the paragraph on page 5, under “Data extraction and criteria”:

1. Referring instead to diagnostic tests requested following the check
2. Adding a sentence to make clear that CVD diagnoses and medication prescription data will be presented in subsequent papers. As this is the first of a series of papers, we have sought to ensure that the methods and details of the data extract are detailed fully here, so that this paper can be referenced subsequently for all later analyses.

“The data extracted for each individual included socio-demographic characteristics, risk factors for cardiovascular disease, diagnostic tests requested following the check, and interventions including advice and referrals. CVD diagnoses and medication data were also extracted from three out of the four GP clinical IT systems providers, corresponding to 60% of practices. Data extraction for all variables were restricted to time windows around the individual’s contact with the NHSHC programme as specified in the business rules for extraction, listed in Supplementary Table 2. Data for CVD diagnoses and a broader range of medications will be presented in subsequent papers.”

With regards to the timing of variable capture, we think the complex rules of the time windows for each variable for attendees and non-attendees are best placed in the supplementary material. However if the editorial team would like us to move this table to the main text we would be happy to do so.

While it is true to say that “individuals with current statin prescriptions would not be eligible for an invitation to the NHSHC” it is entirely possible that despite this, some may still have been invited. Data extraction does not seem to take account of this possibility, so is there an assumption that any statin prescription is newly initiated and is this a valid assumption?

Thank you for raising this point. We only estimated new prescriptions, defined by date of first prescription among those with no prior record of statin prescribing. This has been clarified in the definitions of study variables under methods on page 6.

“Statin prescriptions that occurred on or after the index data among attendees with no data for previous statin prescription were selected.”

In the results this is further reinforced on page 11, under the section on statin prescriptions:

“Information on a new statin prescription, occurring on or after NHSHC completion.....”

Results

Page 11 It is not entirely correct to that that “Completeness of all risk factors was lower among non-attendees” as more non-attenders had HbA1c recorded. There were similar proportions with records of fasting glucose and recording of smoking and blood pressure are also pretty high in non-attenders. Suggest modify Discussion slightly to reflect this.

We thank the reviewer for highlighting this. Generally most core risk factors like smoking were better recorded in attendees (eg smoking 24% missing in non-attendees v 4% in attendees). However as the reviewer correctly points out, recording of HbA1c was higher in non-attendees and similar for fasting glucose. We have edited the sentence in the results section, line 10, page 11 to read:

“Completeness of most, but not all risk factors, was lower among non-attendees, with the exception of diabetes risk measurements which were similarly low in both groups.”

And we explore the reasons for this in the discussion, paragraph 3, page 13:

“This contrasts with the high degree of missing data among non-attendees for most risk factors. The exception being blood glucose and HbA1C measurements which were similarly complete at low levels for both non-attendees and attendees. This may be because these tests are only performed in attendees at high diabetes risk, combined with parallel current or historical efforts to establish and maintain a diabetes disease register outside of the NSHC.”

Page 12 “Advice, information and referral recorded almost six million times for all attendees, and more than 2.5 million times for individuals with elevated CVD risk factors.” What proportion of those with high risk score were given advice compared to those with <10% risk score? What was the actual number/proportion of attendees given any advice and the number with one, two, three pieces of advice/referral? (Potentially useful as the more people are asked to change, the less they are likely to do so.)

Thank you for raising this point. As risk factors and interventions are the major focus of the next paper in this series of papers, we have not expanded on this here given this paper is an overview with a focus on delivery and uptake of the NSHC. A broad high-level description of the interventions is provided to allow comparison with similar NSHC papers.

Table 3 Referrals to DPP are given, but isn't it the case that this had not been launched at the start of this data extraction period? No comment in Discussion on the small number of such referrals.

The reviewer is correct that the diabetes prevention programme (DPP) started after our data extraction start period and we therefore only capture the first two years of the DPP while it was being mobilised. The low numbers reflect this but also because documentation of referral to the program was poorly recorded during this time. <https://onlinelibrary.wiley.com/doi/full/10.1111/dme.13562>
We have added this point to the discussion on page 13, line 16.

“We note however that these figures may be an underestimate being entirely dependent on coding practices and availability of services by region. For example, the low referral rates for the diabetes prevention programme (DPP) are partly explained by the programme launching relatively recently in 2016, but also due to variation in its availability across England and the poor recording of referrals to the programme in the primary care record as reported by others.²⁹”

The proportion in the general practices from which prescribing data was extracted who had a 10% CVD risk score or above was 21.6%, slightly higher than for full sample (20.4%). This is not stated. We have corrected these numbers as described above. Rather than report the number who had a 10 year risk score >10% in the whole attendee population (irrespective of missing data), we now report this from the proportion in whom the CVD risk score was numerically reported, which is 25.9%. The equivalent proportion from those with a documented CVD risk score and prescribing data was 25.8%. As this is almost identical, we have not discussed the difference in detail.

Key for Figs 1 and 2 (especially Fig 1) could be more useful – it's quite hard to see what % the different colours represent. Why not include all reported levels in both?

Thank you for highlighting this. We assume this refers to figures 2a and 2b as figure 1 is the study flow chart. We have edited the figure key so that more of the scale is visible. We also highlight that data for attendance for each UTLA is available in supplementary table 9.

Is Fig 3 format the best way of showing these data?

This particular format was agreed upon by the writing group as displaying the differences in a visually clear way and not adding more bar charts to improve presentation and readability of the paper.

Readers can find the exact numbers for completeness of risk factor measurements in Supplementary Table 12. We now mention this in the figure legend.

“See also Supplementary Table 12 for completeness values.”

Fig 4 may be better if each risk factor was presented for both attendees and non-attendees together, enabling easier comparisons. Perhaps consider horizontal bar chart.

We thank the reviewer for raising this. There were extensive discussions among the writing group about this figure. It was agreed that due to differing missingness rates (likely not at random), direct comparison of risk factor prevalence for attendees and non-attendees was not meaningful and discouraged. These data are therefore presented separately, but the risk factors are presented in the same order to facilitate interpretation. Readers can find the raw numbers in Supplement Table 6 and we mention this in the figure legend.

“See also Supplementary Table 6 for more detailed information.”

Discussion

A CVD risk score was formally recorded for 79.7% of attendees. In theory it should be 100%. No comment is made on whether this is considered a good or a poor result.

Added a comment

We thank the reviewer and agree all practitioners should have calculated a 10 year risk score as part of the NHSHC. However, we have identified that some providers may have used separate tools to calculate the 10 year risk (eg online QRISK or JBS3 calculator), which was not integrated in the clinical template and values may not have been imported directly. The score may therefore still have been calculated and used in the discussion with the patient. However we have no way of confirming this and some practitioners may not have used the tools at all. We highlight this now in discussion on page 14, line 1:

“A 10 year risk score was documented in 79.7% of all attendees. We anticipate that in the remaining ~20%, practitioners may have estimated the score using an online or other tool not integrated into the clinical system, which may have meant the score was calculated but not recorded, although it is possible some may not have calculated it at all. Overall, where a score was recorded a quarter of all attendees were calculated to have a 10-year CVD risk score of $\geq 10\%$,.....”

“The programme appears to have been successful at promoting advice and guideline-based interventions.” While it is true to say that the extent or quality of the conversations around risk or the suitability of information given is not known, it is also a limitation that no data are reported on whether advice was followed, referrals accepted etc and actual clinical outcomes or even proxy outcomes (reduced BP, cholesterol, weight). Such outcomes are not part of this work (and may not be recorded). While this seems obvious, it may be useful to just add this point to the text.

We have amended the final sentence in the clinical implications to address this issue and point to the fact that future analyses will explore the outcomes in more detail.

“Although assessing the efficacy of these interventions on individual level behaviour change is challenging, further analysis of this large dataset will explore the impact on available metrics such as diagnosis rates and clinical outcomes”

Reviewer: 2

Thank you for the opportunity to review this interesting manuscript.

I do not have any substantial comments. The background and methods sections are clear and well-written, and the authors present a balanced discussion of the results and the limitations due to e.g. missing values. The results are very interesting, especially given the large dataset and availability of information.

We thank the reviewer for these positive comments